# Separability Criteria Based on the Weyl Operators

**DOI:** 10.3390/e24081064

**Published:** 2022-08-02

**Authors:** Xiaofen Huang, Tinggui Zhang, Ming-Jing Zhao, Naihuan Jing

**Affiliations:** 1School of Mathematics and Statistics, Hainan Normal University, Haikou 571158, China; huangxf1206@163.com; 2Key Laboratory of Data Science and Smart Education, Ministry of Education, Hainan Normal University, Haikou 571158, China; 3School of Science, Beijing Information Science and Technology University, Beijing 100192, China; zhaomingjingde@126.com; 4Department of Mathematics, Shanghai University, Shanghai 200444, China; jing@ncsu.edu; 5Department of Mathematics, North Carolina State University, Raleigh, NC 27695, USA

**Keywords:** separability, Weyl operators, Bloch representation, correlation matrix, quantum teleportation

## Abstract

Entanglement as a vital resource for information processing can be described by special properties of the quantum state. Using the well-known Weyl basis we propose a new Bloch decomposition of the quantum state and study its separability problem. This decomposition enables us to find an alternative characterization of the separability based on the correlation matrix. We show that the criterion is effective in detecting entanglement for the isotropic states, Bell-diagonal states and some PPT entangled states. We also use the Weyl operators to construct an detecting operator for quantum teleportation.

## 1. Introduction

Quantum information processing is responsible for implementing tasks such as super-dense coding [1], teleportation [2] and key generation [3]. Quantum entanglement is one of the key sources [4] of quantum advantages and has many applications ranging from quantum teleportation to quantum cryptography [5]. In recent years, much effort has been devoted to understanding entanglement, but still many problems remain unsolved. One key problem is to determine whether a given bipartite state is entangled or not. Recall that a bipartite quantum state ρ in a Hilbert space $H_{A}⊗H_{B}$ is *separable* if
(1)$$\rho =\sum_{i}p_{i}\rho_{i}^{A}⊗\rho_{i}^{B},$$
where $\left\{p_{i}\right\}$ is a probability distribution, $\rho_{i}^{A}$ and $\rho_{i}^{B}$ are the reduced density matrices of subsystem $H_{A}$ and $H_{B}$ respectively. Otherwise ρ is said to be *entangled*.

For low-dimensional bipartite systems such as 2⊗2, 2⊗3 and 3⊗2 systems, the celebrated PPT criterion [6] is a necessary and sufficient condition for separability. However, for higher-dimensional multipartite systems, entanglement detection is widely believed to be an NP-hard problem. Nevertheless there are several separability criteria available. Among them, a notable one is entanglement witness which detects entanglement theoretically and experimentally [7], and most linear separability criteria can be regarded as entanglement witnesses. As a nonlinear separability criterion the local uncertainty relation [8] is an effective method to detect entanglement, and there are some nonlinear criteria based on matrix methods, for example, the realignment criterion [9], the covariance matrix criterion [10] and the separability criteria based on the correlation matrix [11,12].

In this paper, we focus on an improved Bloch representation of density matrix in terms of the Weyl basis to derive separability criteria. Our work shows that the Weyl basis is advantageous in handling higher dimensional quantum states as well as revealing the symmetry property. This new scheme markedly simplifies calculations involving density matrices. Our method further demonstrates that Weyl operators can be widely applied in quantum information realm [13,14,15]. Particularly, Weyl operators also play important roles in entanglement detection [16,17,18,19,20,21,22,23]. In fact, Ref. [16] provided the generalized Pauli matrices based on the Weyl operators, and proposed a criterion to detect entanglement by the bounds of the sum of expectation values of any set of anti-commuting observables. Moreover, the separability criteria in terms of the Weyl operators for bipartite and multipartite quantum systems were presented in Ref. [17]. The Weyl operators also play a crucial role in constructing the Weyl discrete channels. Furthermore, the Weyl operators have been widely used in representation theory of affine Lie algebras and Yangians [24].

The layout of the paper is as follows. In Section 2 we first show that Weyl operators provide a generalization of the Pauli operators that can represent any quantum state in a tensor format. Based on the Weyl representation of quantum state, we will present a separability criterion in terms of the correlation matrix for an arbitrary bipartite quantum state in Section 3. Our method also gives a necessary and sufficient condition for separability, which is applicable in quantum teleportation, as shown in Section 4. Detailed examples are provided to illustrate the advantages of this method compared with previous methods.

## 2. The Representation of Quantum States in Terms of Weyl Operators

Let H be a *d*-dimensional Hilbert space with computational basis {|k〉}, and $Z_{d}$ denotes the finite field of modulo *d* integers. For simplicity all integers in the subscripts are modulo *d*. Recall that the Weyl operators are defined by
(2)$$W_{nm}:=\sum_{k\in Z_{d}}e^{\frac{2kn\pi i}{d}}|k\rangle \langle \left(k+m\right)\mathrm{mod}\;d|,\;\;n,m=0,1,\cdots ,d-1.$$
Clearly the set $\left\{W_{nm}\right\}$ forms a basis of linear generators in the general linear Lie algebra gl(d). We remark that the Weyl basis is also called the principal basis in the literature. When d=2, the Weyl operators specialize to the Pauli matrices, i.e., $\left\{W_{00},W_{01},W_{10},W_{11}\right\}=\left\{I,\sigma_{1},\sigma_{3},-i\sigma_{2}\right\}$. In general when d=≥2, the Weyl basis is different from both the Cartan–Weyl basis and Gell-Mann basis. The Weyl operators $\left\{W_{nm}\right\}$ enjoy the following algebraic relations
$$W_{ij}W_{kl}=e^{\frac{2jk\pi i}{d}}W_{i+k,j+l},\;\;W_{kl}^{†}=e^{\frac{2kl\pi i}{d}}W_{-k,-l}.$$
Although the Weyl operators are not Hermitian in general, they are unitary and enjoy the orthogonal relation
$$\langle W_{nm},W_{kl}\rangle =\mathrm{Tr}W_{nm}^{†}W_{kl}=d\delta_{nk}\delta_{ml},$$
where $\delta_{ij}$ is the Kronecker symbol. Subsequently the Weyl operators obey the trace relation
$$\mathrm{Tr}W_{ij}=\left\{\begin{array}{ccc} d, & & (i,j)=(0,0),\\ 0, & & \mathrm{otherwise}. \end{array}\right.$$

As an example, there are nine linearly independent Weyl operators on a 3-dimensional Hilbert space listed as follows:$$\begin{array}{ccc} W_{00}=\left(\begin{array}{ccc} 1 & 0 & 0\\ 0 & 1 & 0\\ 0 & 0 & 1 \end{array}\right), & W_{01}=\left(\begin{array}{ccc} 0 & 1 & 0\\ 0 & 0 & 1\\ 1 & 0 & 0 \end{array}\right), & W_{02}=\left(\begin{array}{ccc} 0 & 0 & 1\\ 1 & 0 & 0\\ 0 & 1 & 0 \end{array}\right),\\ W_{10}=\left(\begin{array}{ccc} 1 & 0 & 0\\ 0 & \omega & 0\\ 0 & 0 & \omega^{2} \end{array}\right), & W_{11}=\left(\begin{array}{ccc} 0 & 1 & 0\\ 0 & 0 & \omega \\ \omega^{2} & 0 & 0 \end{array}\right), & W_{12}=\left(\begin{array}{ccc} 0 & 0 & 1\\ \omega & 0 & 0\\ 0 & \omega^{2} & 0 \end{array}\right),\\ W_{20}=\left(\begin{array}{ccc} 1 & 0 & 0\\ 0 & \omega^{2} & 0\\ 0 & 0 & \omega \end{array}\right), & W_{21}=\left(\begin{array}{ccc} 0 & 1 & 0\\ 0 & 0 & \omega^{2}\\ \omega & 0 & 0 \end{array}\right), & W_{22}=\left(\begin{array}{ccc} 0 & 0 & 1\\ \omega^{2} & 0 & 0\\ 0 & \omega & 0 \end{array}\right), \end{array}$$
where ω is a 3rd primitive unit root of 1, i.e., $\omega^{3}=1$.

Since the $d^{2}$ linearly independent Weyl operators $W_{nm}$ form a basis of gl(d), every d×d density matrix ρ can be uniquely expressed as a linear combination of the Weyl basis:(3)$$\rho =\frac{1}{d}\left(I+\sum_{\left(i,j)\neq (0,0\right)}a_{ij}W_{ij}\right),$$
where the coefficients $a_{ij}=\mathrm{Tr}W_{ij}^{†}\rho$ for i,j=0,1,⋯,d−1. Since $\rho^{†}=\rho$, the coefficients satisfy the symmetry condition
(4)$$a_{nm}^{*}=e^{\frac{-2nm\pi i}{d}}a_{-n,-m},$$
where * means complex conjugation. We also call $\nu =(a_{ij})$ the Bloch vector of ρ relative to the Weyl basis and its length is defined as $\left|\nu \right|=\sum_{i,j}{\left|a_{ij}\right|}^{2}$. Therefore any density matrix ρ in Hilbert space H can be uniquely characterized by a $d^{2}-1$ dimensional vector $\nu \in C^{d^{2}-1}$ with the symmetry condition (Equation 4), that is, $d^{2}-1$ real parameters.

**Theorem** **1.**
*For any d-dimensional quantum state ρ in the form of Equation (Equation 3), the length of the vector ν=ν(ρ) satisfies the following inequality*

(5)
$$\left|\nu \right|\leq \sqrt{d-1}.$$

*In particular, the equality holds if and only if ρ is pure.*


**Proof.** Since any quantum state ρ satisfies the trace condition $\mathrm{Tr}\rho^{2}\leq 1$, one obtains that
$$\begin{array}{ccc} \mathrm{Tr}\rho^{2} & = & \frac{1}{d^{2}}\mathrm{Tr}\left(I+\sum a_{ij}W_{ij}\right)\left(I+\sum a_{kl}W_{kl}\right)\\ & = & \frac{1}{d}+\frac{1}{d^{2}}\sum e^{\frac{2jk\pi i}{d}}a_{ij}a_{kl}\mathrm{Tr}W_{i+k,j+l}\leq 1. \end{array}$$
Note that $W_{00}=I$ and the matrices $W_{ij}$ are traceless for (i,j)≠(0,0), the only nonzero terms in the summation are for k=−i and l=−j, that is,
(6)$$\mathrm{Tr}\rho^{2}=\frac{1}{d}\left(1+\sum e^{\frac{-2ij\pi i}{d}}a_{ij}a_{-i,-j}\right)\leq 1.$$
Using the symmetry condition (Equation 4), the trace is simplified as
(7)$$\mathrm{Tr}\rho^{2}=\frac{1}{d}\left(1+\sum a_{ij}a_{ij}^{*}\right)=\frac{1}{d}{\left(1+|\nu \right|}^{2})\leq 1.$$
Therefore $\left|\nu \right|\leq \sqrt{d-1}$. If ρ is pure, we have $\mathrm{Tr}\rho^{2}=1$, which means $\left|\nu \right|=\sqrt{d-1}$. This completes the proof. □

Theorem 1 tells us that all Bloch vectors lie within a hypersphere of radius $\sqrt{d-1}$ with the pure states on the spherical surface. Moreover, the quantum state ρ is determined by the Bloch vector ν=ν(ρ) satisfying ν with the symmetry condition (Equation 4), thus the set of the Bloch vectors is a subset of the vector space $C^{d^{2}-1}$ with $d^{2}-1$ parameters.

## 3. Application of Weyl Operators in Separability Criteria

With the Weyl basis, a quantum state ρ in space $H_{A}⊗H_{B}$ with $dimH_{A}=d_{A}$ and $dimH_{B}=d_{B}$ can be decomposed as
(8)$$\rho =\frac{1}{d_{A}d_{B}}\left(I_{A}⊗I_{B}+\sum_{\left(i,j)\neq (0,0\right)}\alpha_{ij}W_{ij}^{A}⊗I_{B}+\sum_{\left(k,l)\neq (0,0\right)}\beta_{kl}I_{A}⊗W_{kl}^{B}+\sum_{\left(i,j),(k,l)\neq (0,0\right)}\lambda_{i,j}^{k,l}W_{ij}^{A}⊗W_{kl}^{B}\right),$$
where the coefficients $\alpha_{ij}=\mathrm{Tr}\rho {\left(W_{ij}^{A}\right)}^{†}⊗I_{B}$, $\beta_{kl}=\mathrm{Tr}\rho I_{A}⊗{\left(W_{kl}^{B}\right)}^{†}$, $\lambda_{i,j}^{k,l}=\mathrm{Tr}\rho {\left(W_{ij}^{A}\right)}^{†}⊗{\left(W_{kl}^{B}\right)}^{†}$, $I_{A(B)}$ and $W_{ij}^{A(B)}$ are the identity operator and the Weyl operators of space $H_{A(B)}$, respectively. Let α and β be two complex vectors with dimension $d_{A}^{2}-1$ and $d_{B}^{2}-1$, respectively, that is,
$$\begin{array}{c} \alpha =\left(\alpha_{ij}\right)={\left(\alpha_{01},\ldots ,\alpha_{0,d_{A}-1},\ldots ,\alpha_{d_{A}-1,0},\ldots ,\alpha_{d_{A}-1,d_{A}-1}\right)}^{t},\\ \beta =\left(\beta_{kl}\right)={\left(\beta_{01},\ldots ,\beta_{0,d_{B}-1},\ldots ,\beta_{d_{B}-1,0},\ldots ,\beta_{d_{B}-1,d_{B}-1}\right)}^{t}, \end{array}$$
where *t* denotes transposition. The entries $\lambda_{i,j}^{k,l}$ form a matrix *M* with size $\left(d_{A}^{2}-1\right)\times \left(d_{B}^{2}-1\right)$, which will be referred to as the *correlation matrix* of ρ (relative to the Weyl bases). It is the correlation matrix of ρ under the Weyl basis. Since $\rho^{†}=\rho$, the entries satisfy the symmetry conditions: $\alpha_{ij}^{*}=e^{\frac{-2ij\pi i}{d_{A}}}\alpha_{-i,-j}$, $\beta_{kl}^{*}=e^{\frac{-2kl\pi i}{d_{B}}}\beta_{-k,-l}$, ${\left(\lambda_{i,j}^{k,l}\right)}^{*}=e^{-2\pi i(\frac{ij}{d_{A}}+\frac{kl}{d_{B}})}\lambda_{-i,-j}^{-k,-l}$.

According to the decomposition in Equation (Equation 8), the reduced states of ρ on the two subsystems are respectively given by
(9)$$\rho_{A}={\mathrm{Tr}}_{B}\rho =\frac{1}{d_{A}}\left(I_{A}+\sum_{\left(i,j)\neq (0,0\right)}\alpha_{ij}W_{ij}^{A}\right),$$
(10)$$\rho_{B}={\mathrm{Tr}}_{A}\rho =\frac{1}{d_{B}}\left(I_{B}+\sum_{\left(k,l)\neq (0,0\right)}\beta_{kl}W_{kl}^{B}\right).$$

**Theorem** **2.**
*A bipartite quantum state ρ in the form of Equation (Equation 8) is a product state, i.e., $\rho =\rho_{A}⊗\rho_{B}$, if and only if the correlation matrix M is of rank 1: the matrix M can be written as*

(11)
$$M=\alpha \beta^{t}.$$

*for some column vectors α and β.*


**Proof.** One notices that Equation (Equation 8) can be rewritten as
(12)$$\rho =\rho_{A}⊗\rho_{B}+\frac{1}{d_{A}d_{B}}\sum_{\left(i,j),(k,l)\neq (0,0\right)}\left(\lambda_{i,j}^{k,l}-\alpha_{ij}\beta_{kl}\right)W_{ij}^{A}⊗W_{kl}^{B}.$$
Since the matrices $W_{ij}^{A}⊗W_{kl}^{B}$ are linearly independent, $\left(\lambda_{i,j}^{k,l}-\alpha_{ij}\beta_{kl}\right)W_{ij}^{A}⊗W_{kl}^{B}=0$ if and only if $\lambda_{i,j}^{k,l}-\alpha_{i,j}\beta_{k,l}=0$ for (ij),(kl)≠(0,0), that is, $M=\alpha \beta^{t}$, and this completes the proof. □

Since any mixed state is a convex combination of pure states, Theorem 2 provides a necessary condition for separability for any mixed state in a bipartite system.

Now we denote the Ky Fan matrix norm of *M* as ${∥M∥}_{KF}=\sum \xi_{i}=\mathrm{Tr}\sqrt{M^{†}M}$, which is the sum of the singular values $\xi_{i}$ of the matrix *M*. Notice that the Ky Fan norm, the trace norm and the Shatten-1 norm are uniform for a square matrix. Then we have the following necessary condition for separability for any bipartite quantum state.

**Theorem** **3.**
*If a bipartite state ρ in the form of Equation (Equation 8) is separable, then the correlation matrix M of ρ satisfies the inequality as follows*

(13)
$${∥M∥}_{KF}\leq \sqrt{\left(d_{A}-1\right)\left(d_{B}-1\right)}.$$



**Proof.** Suppose the quantum state ρ is separable, then there exists a series of $p_{s}$, $\rho_{s}^{A}$, $\rho_{s}^{B}$ such that $\rho =\sum_{s}p_{s}\rho_{s}^{A}⊗\rho_{s}^{B}$, with $p_{s}\geq 0$ and $\sum_{s}p_{s}=1$. Suppose $\rho_{s}^{A}=\frac{1}{d_{A}}\left(I_{A}+\sum_{\left(i,j)\neq (0,0\right)}\alpha_{ij}^{\left(s\right)}W_{ij}^{A}\right)$ and $\rho_{s}^{B}=\frac{1}{d_{B}}\left(I_{B}+\sum_{\left(k,l)\neq (0,0\right)}\beta_{kl}^{\left(s\right)}W_{kl}^{B}\right)$ with Bloch vectors $\alpha_{s}=\left(\alpha_{ij}^{\left(s\right)}\right)$ and $\beta_{s}=\left(\beta_{ij}^{\left(s\right)}\right)$, respectively. Therefore the correlation matrix *M* of ρ is $M=\sum_{s}p_{s}\alpha_{s}\beta_{s}^{t}$. One sees that
(14)$$\begin{array}{ccc} {∥M∥}_{KF} & \leq & \sum_{s}p_{s}∥\alpha_{s}\beta_{s}^{t}{∥}_{KF}=\sum_{s}p_{s}|\alpha_{s}\left|\right|\beta_{s}|\\ & \leq & \sum_{s}p_{s}\cdot \sqrt{d_{A}-1}\cdot \sqrt{d_{B}-1}=\sqrt{\left(d_{A}-1\right)\left(d_{B}-1\right)}. \end{array}$$
where we have used the equality of norm [25], for any pure state |a〉 and |b〉,
(15)$${∥|a\rangle \langle b|∥}_{KF}=||a\rangle |||b\rangle |.$$
This completes the proof. □

**Example** **1.**
*Consider the isotropic state $\rho_{iso}=\left(\frac{1-p}{d^{2}}\right)I⊗I+p|\psi_{+}\rangle \langle \psi_{+}|$, where 0≤p≤1. They are separable if and only if $p\leq \frac{1}{d+1}$ [26]. Since the maximally-entangled pure state $|\psi_{+}\rangle =\frac{1}{\sqrt{d}}\sum_{i=0}^{d-1}|ii\rangle$ has Bloch decomposition relative to Weyl operators as follows*

(16)
$$|\psi_{+}\rangle \langle \psi_{+}|=\frac{1}{d^{2}}I⊗I+\sum_{\left(i,j)\neq (0,0\right)}\frac{1}{d^{2}}W_{ij}⊗W_{-i,-j}.$$

*Therefore, the isotropic state can be represented as*

(17)
$$\rho_{iso}=\frac{1}{d^{2}}\left(I⊗I+\sum_{\left(i,j)\neq (0,0\right)}pW_{ij}⊗W_{-i,-j}\right).$$

*Note that the Ky Fan norm of correlation matrix M of $\rho_{iso}$ is ${∥M∥}_{KF}=\left(d^{2}-1\right)p$. Theorem 3 implies that $p\leq \frac{1}{d+1}$ when the isotropic state is separable. This means that we can detect all entangled isotropic states by Theorem 3.*


**Example** **2.**
*Let ρ be the following 3×3 PPT entangled state found in [27]:*

(18)
$$\rho =\frac{1}{4}(I-\sum_{i=0}^{4}|\chi_{i}\rangle \langle \chi_{i}|),$$

*where $|\chi_{0}\rangle =|0\rangle (|0\rangle -|1\rangle )/\sqrt{2}$, $|\chi_{1}\rangle =(|0\rangle -|1\rangle )|2\rangle /\sqrt{2}$, $|\chi_{2}\rangle =|2\rangle (|1\rangle -|2\rangle )/\sqrt{2}$, $|\chi_{3}\rangle =(|1\rangle -|2\rangle )|0\rangle /\sqrt{2}$, $|\chi_{4}\rangle =(|0\rangle +|1\rangle +|2\rangle )(|0\rangle +|1\rangle +|2\rangle )/3$. We get the Ky Fan norm of the correlation matrix ${∥M∥}_{KF}$ approximately equal to 2.15, which violates the inequality in Theorem 3. Therefore the state ρ is entangled.*


**Example** **3.**
*The Bell-diagonal states can be represented as $\rho =\frac{1}{4}\left(I_{4}+\sum_{i=1}^{3}t_{i}\sigma_{i}⊗\sigma_{i}\right)$, where $\sigma_{i}$ are three Pauli operators [28]. The Bell-diagonal states are known to be separable iff $|t_{1}\left|+\right|t_{2}\left|+\right|t_{3}|\leq 1$ [28]. Consider the Bloch decomposition of ρ relative to the Weyl operators*

(19)
$$\rho =\frac{1}{4}\left(I_{2}⊗I_{2}+t_{1}W_{01}⊗W_{01}+t_{3}W_{10}⊗W_{10}-t_{2}W_{11}⊗W_{11}\right).$$

*Then the Ky Fan norm of the correlation matrix is ${∥M∥}_{KF}=|t_{1}\left|+\right|t_{2}\left|+\right|t_{3}|$. It follows from Theorem 3 that $|t_{1}\left|+\right|t_{2}\left|+\right|t_{3}|\leq 1$ when the Bell-diagonal states are separable. Again Theorem 3 completely detects the entanglement for the Bell-diagonal states.*


**Example** **4.**
*Consider the bipartite state*

(20)
$$\rho_{\pm }=p|\phi_{\pm }\rangle \langle \phi_{\pm }|+\left(1-p\right)|00\rangle \langle 00|,$$

*where p∈[0,1], and $|\phi_{\pm }\rangle =\frac{1}{\sqrt{2}}(|01\rangle \pm \left|10\rangle \right)$. The Peres–Horodecki criterion establishes that the state is separable iff p=0 [6]. For the Bloch representation using Weyl operators, thus we obtain that*

(21)
$$\begin{array}{ccc} \rho_{\pm } & = & \frac{1}{4}(I_{2}⊗I_{2}+\left(1-p\right)W_{10}⊗I_{2}+\left(1-p\right)I_{2}⊗W_{10}\pm pW_{01}⊗W_{01}\\ & & \mp pW_{11}⊗W_{11}+\left(1-2p\right)W_{10}⊗W_{10}). \end{array}$$

*Therefore, the Ky Fan norm of the correlation matrix M of state $\rho_{\pm }$ is ${∥M∥}_{KF}=2p+\left|1-2p\right|$, which implies that ${∥M∥}_{KF}\leq 1$ if $p\leq \frac{1}{2}$, so entanglement is detected only if $p>\frac{1}{2}$.*


## 4. Application of Weyl Operators in Quantum Teleportation

In the process of quantum teleportation, the optimal fidelity of teleportation as an entangled resource can be expressed by fully entangled fraction [29,30,31]. For a given quantum state ρ in *d*-dimensional Hilbert space, the optimal fidelity of telepotation with respective to ρ can be described by the function
(22)$$f_{max}\left(\rho \right)=\frac{dF(\rho )}{d+1}+\frac{1}{d+1},$$
where F(ρ) is the fully entangled fraction with respect to ρ defined by [30]:(23)$$F\left(\rho \right)=\max_{U}\langle \psi_{+}|\left(U^{†}⊗I\right)\rho \left(U⊗I\right)|\psi_{+}\rangle ,$$
where *U* runs through all d×d unitary matrices, *I* is the d×d identity matrix, and $|\psi_{+}\rangle$ is the maximally entangled state. A state ρ is a useful resource for teleportation if and only if $F\left(\rho \right)>\frac{1}{d}$ [30]. If $F\left(\rho \right)\leq \frac{1}{d}$, the fidelity is considered to be no better than separability. In this sense, the fully entangled fraction F(ρ) can be used to detect a quantum teleportation resource. Ref. [32] gave an elegant formula for a two-qubit system by using the method of Lagrange multipliers. Refs. [33,34] constructed the teleportation witness for detecting the quantum states that are useful for quantum teleportation.

Now we construct an operator using the Weyl representation to detect if a quantum state is useful for quantum teleportation. Since the maximally entangled state $|\psi_{+}\rangle =\sum_{i}\frac{1}{\sqrt{d}}|ii\rangle$ can be decomposed as Equation (Equation 16) according to the Weyl operators [35], we let $P_{ij}=UW_{ij}U_{U}^{†}$ and define a normal operator O by
(24)$$\begin{array}{c} O_{U}:=I⊗I+\sum_{\left(i,j)\neq (0,0\right)}P_{ij}⊗W_{-i,-j}. \end{array}$$
We claim that the operator $O_{U}$ can be used to detect whether an unknown quantum state is available for teleportation.

**Theorem** **4.**
*The quantum state ρ in a quantum system H⊗H with dim(H)=d is useful for teleportation if and only if there exists some unitary operator U such that the mean value satisfies the inequality:*

(25)
$${\langle O_{U}\rangle }_{\rho }>d.$$



**Proof.** For any quantum state ρ, it has
$$\begin{array}{ccc} \langle O_{U}\rangle_{\rho } & = & \langle I⊗I+\sum_{\left(i,j)\neq (0,0\right)}P_{ij}⊗W_{-i,-j}\rangle_{\rho }\\ & = & \langle I⊗I+\sum_{\left(i,j)\neq (0,0\right)}UW_{ij}U^{†}⊗W_{-i,-j}\rangle_{\rho }\\ & = & d^{2}\langle U⊗I|\psi_{+}\rangle {\langle \psi_{+}|U^{†}⊗I\rangle }_{\rho } \end{array}$$Since $\max_{U}{\langle O_{U}\rangle }_{\rho }=d^{2}F\left(\rho \right)$, and quantum state is useful for quantum teleportation if and only if $F\left(\rho \right)>\frac{1}{d}$. Note that the maximum value is attainable since SU(d) is compact. Therefore the quantum state is useful for quantum teleportation if and only if there exists some unitary *U* such that ${\langle O_{U}\rangle }_{\rho }>d$. This completes the proof. □

**Example** **5.**
*Consider the following bipartite state [36]*

(26)
$$\rho =p|\phi^{-}\rangle \langle \phi^{-}|+\left(1-p\right)|00\rangle \langle 00|,$$

*where p∈[0,1] and $|\phi^{-}\rangle =\frac{1}{\sqrt{2}}(|01\rangle -|10\rangle ).$ The PPT criterion establishes that state (Equation 26) is separable iff p=0 [6]. We choose the operator $U=\left|0\rangle \langle 1\right|+\left|1\rangle \langle 0\right|=\sigma_{1}$, then one has ${\langle O\rangle }_{\rho }=3p$. Therefore the quantum state ρ in Equation (Equation 26) is useful for quantum teleportation when $p>\frac{2}{3}$.*


## 5. Conclusions

We have investigated the Bloch decomposition of a density matrix relative to the Weyl basis in the quantum system. The geometric properties of Bloch vectors, including the length, are described in detail. For the bipartite quantum states, we have provided a necessary condition of separability in terms of the Ky Fan norm of the correlation matrix. Furthermore, we have demonstrated the feasibility and effectiveness of our separability criterion in detecting entanglement using examples of isotropic states, Bell-diagonal states and some PPT entangled states. Finally, we have constructed an operator based on the Weyl operators for detecting useful resources for quantum teleportation.

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
