# Peer review of "Separability Criteria Based on the Weyl Operators"

_entropy, 2022, doi:10.3390/e24081064_

Round 1

Reviewer 1 Report

In this paper the authors investigate a Bloch vector description of multipartite quantum states in terms of a Weyl operator basis. Because of the convenient relations obtained when multiplying these Weyl operators together, they are able to show with several theorems that this can be a useful description. The first theorem defines a Bloch hypersphere for this representation. The second gives a necessary and sufficient condition for product states. The third gives a necessary condition for separability. And the fourth gives a condition signifying usefuleness in teleportation tasks. They also consider several interesting examples.

Their main result, Theorem 3, seems to me interesting, simple, and useful - a nice addition to the literature!

Although Bloch vectors based on the Weyl operators have appeared before in the literature in several places, both by the present authors and others, I think the present paper contributes new results and is suitable for publication in Entropy.

Still there are some minor areas which should be addressed by the authors before publication: 

 - Given that the main result is a necessary-but-not-sufficient separability criterion, there ought to be at least some discussion of how faithful this criterion is. In particular - can the authors give an example of an entangled state which nonetheless satisfies the inequality? Are there many such examples? Or are no such states known? This seems like an important piece to gain a full understanding of the Thm. 3.

 - Is Thm. 2 really specific to pure states? If so, why? It looks like the proof holds just fine whether the state is pure or mixed, so long as it is a product. 

 - In the statement of Thm. 3 it would be good to explicitly state that M is the correlation matrix. Although it's fairly apparent after looking at the proof or at the previous theorem, a reader quickly looking to identify the results may be confused (which I was on the first skim-through).

 - Readers may benefit from slightly more explanation in the proof of Thm. 3, particularly near the end, in going from the KyFan norm to the vector norm. I believe it is implied that the authors here have simply evaluated the singular value of this rank-one state; but it will save readers time to state the reasoning used directly.

 - Some readers may be more familiar with this Ky Fan norm as the Trace norm or Schatten-1 norm, which could be worth mentioning.

 - It would be helpful to provide more background references about the use of these Weyl operators/ Weyl Bloch vectors in quantum mechanics, particularly in the introduction, to give some more historical/mathematical context.

 - It would also be helpful to provide more citations to the related literature of Weyl operator Bloch vectors in entanglement, such as arxiv:0806.1174, arxiv:1512.05640, which include some closely related results to this work.

 - The English language and style of the abstract and introduction are somewhat lacking. I think it is fine and does not detract from the results. However, improving the style a bit would probably attract more readers to the paper.

 - This one might be only an error in rendering the draft pdf, I'm not sure. But, in and around Eq.(5) and Thm. 1, the typesetting used to signify a vector (which appears to be an overstar with nothing under it?) is very strange. Probably a more standard notation would help.

Overall I think this is a nice result and paper, and once the authors have appropriately addressed the above I will be happy to recommend publication.

Reviewer 2 Report

Separability Criteria based on the Weyl Operators

by X Huang, T Zhang, M-J Zhao, N Jing

The study is focused on the Bloch representation of the density matrix in terms of the Weyl basis to derive conditions for separability criteria using the a novel correlation matrix. Authors claim that the Weyl basis is advantageous in handling higher dimensional quantum states exploiting symmetry properties. Acording to the study this new scheme simplifies calculations involving density matrices. Furthermore, the method exemplifies Weyl operators to be useful in wide applications of quantum information.

 Arguments presented in the manuscript are convincing enough by showing that Weyl operators might be representd as a basis constituing a generalization of the Pauli operators to specify quantum states in tensor format. Based on the this representation of quantum states through Weyl operators a separability criterion is discussed with respect to the correlation matrix for a bipartite quantum state. This method is shown to provide necessary and sufficient

conditions for separability for quantum teleportation purposes. Finally, some examples are presented to illustrate the method to detect entanglement. 

I consider that the study shows scientific merit in the growing field of quantum information, particularly in helping to develop theoretical tools to detecting entanglement and separability criteria. The study is organized, adequately written, and hence worth considering for publication. My only criticism is on the english writing, it requires major improvement by an expert in the language editing.

Author Response

We thank the referee very much for appreciating our results. We have revised the English expression.